# Unraveling Seasonal Allocation of Soluble Sugars, Starch and Proline in *Sternbergia lutea*

**DOI:** 10.3390/plants12173043

**Published:** 2023-08-24

**Authors:** John Pouris, Evgenia Tampiziva, Sophia Rhizopoulou

**Affiliations:** 1Section of Botany, Department of Biology, National and Kapodistrian University of Athens, Panepistimiopolis, 15784 Athens, Greece; srhizop@biol.uoa.gr; 2Department of Mathematics, National and Kapodistrian University of Athens, 15784 Athens, Greece; tabiziva@aueb.gr

**Keywords:** geophyte, Mediterranean, seasonality, *Sternbergia lutea*

## Abstract

This study focuses on *Sternbergia lutea* (L.) Ker Gawl. ex Spreng., a bulbous, perennial, autumnal flowering geophyte mainly distributed around the Mediterranean Basin. The seasonal content of total sugars, starch and proline in above- and below-ground plant parts in this study, which has hitherto not been the subject of a published study. Geophytes possess underground storage organs that support sprouting, leaf growth and flowering. Furthermore, their buds remain protected below the soil surface during periods of dormancy. Understanding the fluctuation of these compounds in *S. lutea* contributes to our knowledge of its adaptation to the Mediterranean ecosystem. It seems likely that monthly fluctuations in proline accumulation, sugar and starch content in both above- and below-ground tissues of *S. lutea* correspond to the distinct seasonality of the Mediterranean ecosystem. Elevated starch content was investigated in the bulbs, while lower starch content was estimated in the leaves. Substantial values of soluble sugar content have been detected in bulbs and leaves. Additionally, elevated sugar content was detected in the yellow petals of *S. lutea* in October. Pronounced proline accumulation was detected in the leaves and bulbs of *S. lutea* during its active and dormant phase, respectively.

## 1. Introduction

Within the genus *Sternbergia* (Amaryllidaceae) approximately seventeen species are included [1]. The species *Sternbergia lutea* (L.) Ker Gawl. ex Spreng. was placed in the genus *Sternbergia* by Sprengel in 1825 [2]. However, it was first described as *Amaryllis lutea* by Linnaeus in 1753 and as *Oporanthus luteus* (L.) Herb. in 1821. The name of the genus is dedicated to the memory of Kaspar Moritz von Sternberg (1761–1838) of Prague, known as the founder of Paleobotany [3], and the name of the species *lutea* means yellow in Latin. The vernacular names of *Sternbergia lutea* such as autumn daffodil, autumn lily of the field, and yellow autumn crocus, reflect within the context of the season the flowering of this geophyte that is leafless during the hot and dry season and thrives in poor, free-draining soils (Figure 1).

*S. lutea* is native from Spain to West and Central Asia, although its distribution has been expanded to other regions of the western Mediterranean Sea (Figure 2). Additionally, *S. lutea* has been used as an autumnal ornament in private gardens [4,5,6], when other flowering species are scarce. The easily collected bulbs of *S. lutea* (Figure 3), which can be found a few centimeters below the soil surface, and its ornamental traits might have contributed to the transportation and distribution of this geophyte, which has become naturalized in other regions [2,6,7].

Proline fulfils diverse functions in plants and plays a role as a compatible solute under environmental stress conditions. Thus, elevated levels of proline found in plant tissues, functioning as compatible solute, protect cellular structures during dehydration [8]. The accumulation of proline in plant tissues has been associated with response to various abiotic stresses such as drought, salinity, UV irradiation and heavy metal toxicity [9,10,11]. Intercellular proline transport occurs between the cytosol, chloroplasts and mitochondria due to its compartmentalized metabolism. It has been suggested that proline contributes to stabilization of sub-cellular structures, scavenging free radicals and buffering cellular redox potential [12]. Proline metabolism in subcellular compartments contributes to a molecular chaperone stabilizing protein structure, thus protecting cells from stress-induced damage [7,9,13]. Proline is synthesized either in the cytosol or in the chloroplasts from glutamate, which is reduced to glutamate-semialdehyde (GSA) by Δ-1-pyrroline-5-carboxylate synthetase (P5CS). GSA can be converted to pyrroline 5-carboxylate (P5C), which is reduced by P5C reductase (P5CR) to proline [7,14,15]. Free proline accumulation in geophytes varies from species to species and concomitantly to their seasonality [16,17,18,19,20,21,22]. Proline accumulation in Mediterranean plants has mainly been studied in response to a prolonged drought that affects their water status and osmotic potential [11,23,24].

In geophytes, carbohydrates provide carbon and energy for sprouting after a seasonal dormancy, ensuring that the plant has the possibility to develop plant structures, even if the seasonal environmental conditions are not favorable [25,26,27,28].

The objective of the study was to investigate and compare the partitioning of soluble sugars, starch and proline between above- and below-ground parts, as well as in the petals of *Sternbergia lutea*, which are exposed to different environmental conditions. Additionally, the water status of the leaves of *S. lutea* has been investigated. It is noteworthy that *S. lutea* is a synanthous geophyte [16], meaning its leaves and flowers appear simultaneously in October. *S. lutea* is leafless during a 5-month hot and dry, dormant period (from May to September). The study of the above-mentioned ecophysiological traits of *S. lutea* reveals insights into this geophyte’s adaptation to ambient Mediterranean conditions, in Greece.

## 2. Results

### 2.1. Soluble Sugars

The maximum soluble sugar content in the bulbs (Figure 4) of the geophyte *S. lutea* was detected in May (371.03 mg g^−1^), when the visible leaf growth had ceased. During the dry and hot period from June to September, substantially lower values of soluble sugars were detected, and the minimum soluble sugar content in the bulbs was found in August (187.79 mg g^−1^). Relatively low values were detected during winter specifically from December (248.67 mg g^−1^) to February (206.03 mg g^−1^).

Elevated values of soluble sugar content were determined in the new expanding leaves of *S. lutea* in October (259.56 mg g^−1^) and November (251.03 mg g^−1^), while the maximum leaf sugar content was investigated in January (294.85 mg g^−1^). In addition, high sugar content (391.91 mg g^−1^) was estimated in the petals in October (Figure 4).

### 2.2. Starch

Elevated starch content was detected in the bulbs of *S. lutea* in January (379.58 mg g^−1^), February (464.5 mg g^−1^), March (510.75 mg g^−1^), April (758.25 mg g^−1^), May (779.5 mg g^−1^), June (672 mg g^−1^), July (674.53 mg g^−1^), August (657 mg g^−1^) and September (687 mg g^−1^). In October, the starch content in the bulbs increased (717.02 mg g^−1^) concomitantly with the appearance of the new leaves (Figure 5). Subsequently, during the leafy phase and the wet season of the year, the starch content of the bulbs declined to 267.05 mg g^−1^ and 229.50 mg g^−1^ in November and December, respectively.

In October, the starch content in the new leaves of *S. lutea* was 139.52 mg g^−1^ and increased to 300.75 mg g^−1^ in November. The leaf starch content decreased from December (105.75 mg g^−1^) to February (98.25 mg g^−1^). Elevated starch content was noted in March (222.05 mg g^−1^), while the starch content of the leaves declined to 169.50 mg g^−1^ in April (Figure 5). Then, from May to September, *S. lutea* plants grown in the wild are leafless. The leaf starch content of *S. lutea*, in comparison with that of the bulbs, was substantially lower. The starch content in the yellow petals was 75.76 mg g^−1^ in October (Figure 5).

### 2.3. Proline

The seasonal free proline accumulation in the bulbs of *S. lutea* varied from low values in January (9.30 μmol g^−1^) and February (8.64 μmol g^−1^) to elevated values in March (27.50 μmol g^−1^) and April (31.72 μmol g^−1^). A decrease in free proline accumulation in the bulbs of *S. lutea* was observed during the dormant period from May to September (Figure 6). Free proline accumulation was sustained to relatively low values from October (25.91 μmol g^−1^) to December (24.39 μmol g^−1^).

The accumulation of free proline in the new expanding leaves of *S. lutea* was estimated at 20.93 μmol g^−1^ in October and reached a maximum value (39.43 μmol g^−1^) in December. Enhanced proline accumulation was also observed from January to February, while it decreased to 10.21 μmol g^−1^ in the senescent leaves during April (Figure 6). In October, free proline accumulation in the yellow petals of *S. lutea* flowers was found at 20.22 μmol g^−1^, comparable to that of the leaves during the same month (20.93 μmol g^−1^).

### 2.4. Leaf Water Status

In October, the Ψw of young leaves was −0.45 MPa (Table 1), the Ψs was −0.71 MPa, while their turgor (Ψp) was 0.26 MPa. In November and December, Ψw and Ψs were relatively constant (Table 1), while the calculated elevated values of Ψp indicate leaf expansion. The Ψw of leaves decreased from January to April, and lower values of Ψp were calculated (Table 1). The minimum value of turgor was also estimated in April, simultaneously with leaf senescence (Table 1).

## 3. Discussion

Τhe distribution of soluble sugars and starch varies significantly between the leaves and bulbs of *S. lutea* (Figure 4, Figure 5 and Figure 6). In this context, the photosynthetically active leaves i.e., when climatic conditions and sufficient soil water availability in the habitat favor photosynthesis, serve as source from October to April, providing assimilated carbohydrates. These can also be translocated to the bulbs, which are storage organs and serve as sink [28]. The stored carbon reserves in the bulbs support developmental and metabolic requirements during the seasonal sprouting and flowering of the geophyte. It has been argued that sprouting is dependent on sugar dynamics, and carbon source availability is a major determinant of cell division that plays important role in determining phase transitions [26,27].

In the case of *S. lutea*, it is likely that the environmental conditions during autumn favor photosynthetic rates, leading to enhanced soluble sugars in both leaves and bulbs. In October, after the first autumn rains, the hydrolysis of starch in the bulbs may enhance the translocation and use of energy reserves. Also, it has been argued that the natural moisture seems to be sufficient to cause the appearance of synanthus leaves in *S. lutea* [16].

In the bulbs of *S. lutea*, the minimum starch content was estimated after the beginning of its leafy period, but devoid of flowers; subsequently, the starch content increased during leaf expansion to reach its maximum before leaf senescence. Senescence should be considered as an important process in the adaptation of geophytes to environmental conditions [29]. The starch content in the bulbs of *S. lutea* remained elevated and relatively constant during its dormancy, from May to September. Starch is the predominant storage carbohydrate in bulbs [30,31], which starts decreasing during sprouting (dormancy break) in geophytes and is associated with the degradation of the stored form of carbohydrates into translocated and energy-providing derivatives [27]. Elevated starch content in the bulbs during the dormant period of *S. lutea* will provide the required energy reserves for the initiation of sprouting, leaf growth and blossoming in autumn; in other words, as the synanthous *S. lutea* emerges from dormancy, carbohydrates will support its vegetative growth and reproduction [17]. Thus, the bulbs play an important role in storing and providing reserves throughout the plant life cycle. A reduction in the bulb starch content, from November to March, coincides with the leafy phenophase of *S. lutea*.

Comparable values of soluble sugars content were estimated in the leaves of *S. lutea* from October to January, while a decrease was observed from February until the end of the leafy phenophase in April. This reduction may indicate the consumption of soluble sugars, as well as their translocation from the leaves to bulbs (Appendix A), as it has been detected in the beginning of the dormant period (Figure 4), as well as starch degradation (Appendix A). Soluble sugars are the main transported carbohydrates in bulbous plants for either metabolic requirements or functioning of plant organs [19]. Concerning the bulbs of *S. lutea*, a significant correlation was detected between starch and soluble sugar content (Appendix A) during the leafless stage of *S. lutea* (y = 1.5128x − 780.6, R^2^ = 0.8665, *p* < 0.05) and a comparably weaker correlation (y = 0.1328x + 177.27, R^2^ = 0.532, *p* < 0.05) during the leafy stage of *S. lutea* (Appendix A). A correlation between starch and soluble sugar content was not detected for leaves of *S. lutea* (Appendix A). However, leaf soluble sugar content was positively correlated with the ambient temperature (y = 0.0411x + 4.8405, R^2^ = 0.5008, *p* < 0.05). The bulbs are temporary storage organs for carbohydrates, assisting plant growth. These storage organs are sink tissues during the dormant period of the geophyte, while they turn into a source during the growth period. Dormant buds accumulate carbon sources (starch and sugars) to promote growth of the geophyte that adjusts the metabolism between storage and soluble carbohydrates to maintain dormancy, sprouting and a rapid growth during the vegetative phase [26].

The accumulation of free proline in organs of *S. lutea* exhibited a gradual increase in the bulbs during the hot and dry season and elevated values were determined in the leaves during the cold months, which indicate a protective role against the harsh environmental conditions. It was found that free proline accumulation in the leaves of *S. lutea* was negatively correlated with the ambient temperature (y = −0.1594x + 17.813, R^2^ = 0.4262, *p* < 0.05). Also, free proline accumulation in the leaves of *S. lutea* was negatively correlated with that of the bulbs, indicating translocation of proline from leaves to bulbs (y = −0.545x + 34.139, R^2^ = 0.5767, *p* < 0.05) (Appendix A). In addition, free proline accumulation in the bulbs of *S. lutea* was negatively correlated with the monthly precipitation (y = −0.0838x + 69.993, R^2^ = 0.5048, *p* < 0.05) in the research site. Elevated proline content was detected in bulbs of *S. lutea* in August, i.e., during the prolonged drought stress. Under the declining leaf water status of *S. lutea*, soluble sugars and proline contributed to the lowering of the osmotic potential (Ψs), which appeared to be negatively correlated with both soluble sugars (y = −0.0004x − 0.6477, R^2^ = 0.6568, *p* < 0.05) (Appendix A) and free proline (y = 0.0029x − 0.6702, R^2^ = 0.4888, *p* < 0.05) (Appendix A), thus indicating an osmotic adjustment mechanism. The leaf water potential (Ψw) of *S. lutea* appeared to be correlated with both soluble sugars (y = −0.0009x − 0.7031, R^2^ = 0.5688, *p* < 0.05) (Appendix A) and free proline content (y = 0.0064x − 0.6684, R^2^ = 0.5487, *p* < 0.05) (Appendix A). Higher values of Ψw were related to higher soluble sugar and proline content. This could be due to water flow in the leaves, under well-watered conditions during the leafy stage, in the xylem; thereby, elevated translocation of compatible solutes (soluble sugars and proline) may occur via the leaf vascular system, contributing to the maintenance of a favorable gradient for water uptake [32,33]. Enhanced proline accumulation in relation to elevated water potential may also occur when the rate of synthesis exceeds the rate of utilization and/or transportation [34].

It is likely that the seasonal starch and soluble sugar contents in *S. lutea* bulbs support growth and metabolic demands. Autumnal environmental conditions cause dormancy release, sprouting and flowering of the bulbous plants of *S. lutea.* The coordination between source leaves and sink bulbs facilitates the efficient allocation of assimilated carbon, ensuring growth and adaptation of this species to Mediterranean environmental conditions. Free proline accumulation in above- and below-ground organs is also linked to the ability of *S. lutea* to sense and respond to ambient conditions. Nevertheless, further work is required to illustrate traits contributing to the functional adaptation of the geophyte *S. lutea* to the fluctuations of the Mediterranean climate.

## 4. Materials and Methods

### 4.1. Research Site

The study was conducted in plants of *Sternbergia lutea* (L.) Ker Gawl. ex Spreng., growing in the wild, near the campus of the National and Kapodistrian University of Athens, Greece (37°57′58″ N, 23°47′15.8″ E). The active and dormant phases of *S. lutea* were monitored on a monthly basis for two consecutive years (2018 and 2019). Randomly selected plants of *Sternbergia lutea* were observed in their natural habitat, in order to investigate the phenological stages of the plants via detailed field observations. For example the flowers of *S. lutea* appear in October and the leaves emerge from the tuber crown in October. Randomly selected samples of bulbs, leaves and flowers were monthly collected from a single stand of *S. lutea* surrounded by Mediterranean phryganic vegetation, from September 2018 to August 2019; details of this Mediterranean ecosystem and climatic data have been published [19].

### 4.2. Total Sugars and Starch

In order to extract soluble sugars from the samples of *S. lutea* (leaves, bulbs, petals), the plant tissues were dried in a dryer at 70 °C and finely powdered in a grinding mill. These powdered samples were then placed in 10 mL of 80% ethanol (*v*/*v*) and agitated using a shaker. The resulting extracts were filtered using Whatman #2 filter paper. The concentration of soluble sugars was determined through colorimetric analysis, employing a modified phenol–sulfuric acid method [35,36] at a wavelength of 490 nm, using a spectrophotometer (Novaspec III+ Spectrophotometer; Biochrom, Cambridge, UK). After the extraction of sugars, the residue was used for starch determination. The anthrone method [37] was utilized to measure the starch content in the residue; for the colorimetric analysis, a spectrophotometer (Novaspec III+ Spectrophotometer; Biochrom, Cambridge, UK) was employed at a wavelength of 490 nm. D-glucose from Serva (Heidelberg, Germany) was used to prepare aqueous solutions for the standard curve. The results obtained from the analysis are expressed as mg g^−1^ d.w.

### 4.3. Free Proline

The determination of free proline content involved a colorimetric method using 4 mL samples of the condensed fluid extracted from the plant material [38,39]. The extraction procedure and colorimetric determination followed the analytical method that has been previously published [19]. For proline extraction, finely powdered dried tubers, leaves and petals were homogenized with 20 mL of 3% *v*/*v* aqueous sulfosalicylic acid. The homogenate was then filtered through Whatman #2 filter paper. In test tubes, 2 mL of the filtrate was mixed with acid-ninhydrin solution (2 mL) and glacial acetic acid (2 mL). The tubes were placed in a water bath at 100 °C for 1 h and the reaction was terminated by transferring the tubes to an ice bath. After cooling, toluene (4 mL) was added to the reaction mixture, and the mixture was homogenized using a vortex. The chromophore-containing toluene layer was separated from the aqueous phase, and its absorbance was measured at 520 nm using a spectrophotometer. Toluene was used as a blank sample for reference. To determine the proline concentration, a standard curve was created using relevant L-proline solutions (Serva, Heidelberg, Germany). The proline concentration in the samples was calculated on a dry weight basis.

### 4.4. Leaf Water Status

Water potential (Ψ) was assessed using a psychrometric method. Fresh discs with a diameter of 6 mm were cut from leaves of *S. lutea*. The discs were placed in five C-52 psychrometric chambers (Wescor Inc., Logan, UT, USA) connected to a dew point psychrometer (HR-33T, Wescor Inc.) through a psychrometer switchbox (PS-10, Wescor). It took approximately 2 h for the water vapor pressure of the leaf sample to reach equilibrium with that of the psychrometer chamber. The osmotic potential (Ψs) was measured on the same leaf discs, after freezing in liquid nitrogen and thawing, using the above-mentioned technique [40,41,42]. The turgor pressure (Ψp) was calculated as the algebraic difference between Ψ and Ψs. The values of Ψ and Ψs are means of five measurements.

### 4.5. Statistical Analysis

The results are reported as mean ± Standard Error (S.E.). To determine differences in the studied parameters among different plant parts of *S. lutea*, a two-way analysis of variance (ANOVA) was conducted. The significance level was set at *p* < 0.05. If significant differences were observed, post hoc analysis using Duncan’s multiple range test was performed to compare the means. All statistical analyses were carried out using the SPSS statistical software, version 23.0 (SPSS Inc., Chicago, IL, USA). Furthermore, regression analysis was employed to examine relationships between the obtained results of soluble sugars and starch from different plant tissues of *Sternbergia lutea*. This analysis aimed to explore potential associations or dependencies between variables.

## 5. Conclusions

The autumnal sprouting of the geophyte *Sternbergia lutea* reveals the active phase from leaf emergence in October to leaf senescence in April. The dormant phase of *S. lutea* extends from May to September, when the above-ground plant parts are not visible, and therefore they are not exposed to prolonged drought, elevated photoperiods and high temperatures that prevail in the native Mediterranean habitat. The investigated partitioning of soluble sugars, starch and free proline in above- and below-ground parts of *S. lutea* contributes to the understanding of this plant’s ability and metabolic status to maintain its annual rhythm and phenological stages under the Mediterranean environmental conditions. During spring, the increase in starch content in bulbs and leaves of *S. lutea* is coordinated with favorable temperatures for photosynthesis and growth. Starch content in the bulbs was found relatively constant until the later stage of the dormancy of *S. lutea*, while it declined sharply during sprouting. The high soluble sugar content in petals, during the flowering month (October), coincided with elevated values of soluble sugars in leaves and bulbs during the same month (October). The growth cycle of *S. lutea* requires the consumption of reserved carbohydrates in autumn. It seems likely that soluble sugars from bulbs and leaves have been transferred to the yellow petals during the short flowering period of *S. lutea*. In addition, free proline accumulation in the petals was comparable to that of the leaves in October. An increase in free proline accumulation observed in the leaves during winter is most probably related to their exposure to cold stress and osmotic adjustment. The allocation of soluble sugars, starch and proline to above- and below-ground organs of the synanthous *S. lutea* is linked to dormancy, sprouting and flowering synchronized with the environmental conditions.

## Figures and Tables

**Figure 1 plants-12-03043-f001:**
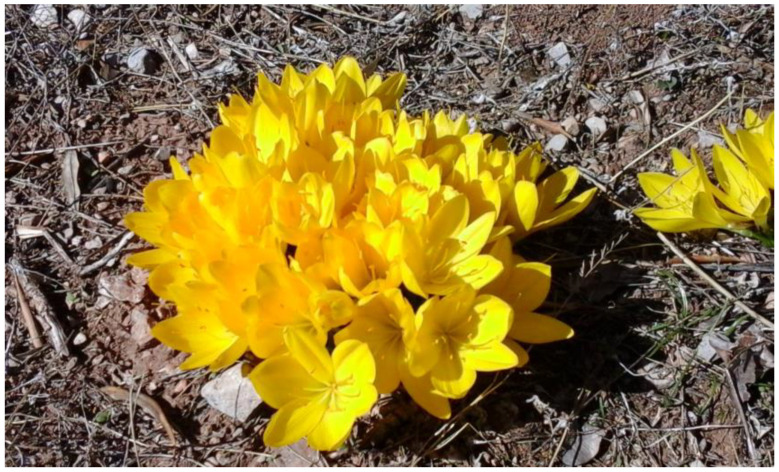
Flowering *Sternbergia lutea* (L.) Ker Gawl. ex Spreng. (© S. Rhizopoulou).

**Figure 2 plants-12-03043-f002:**
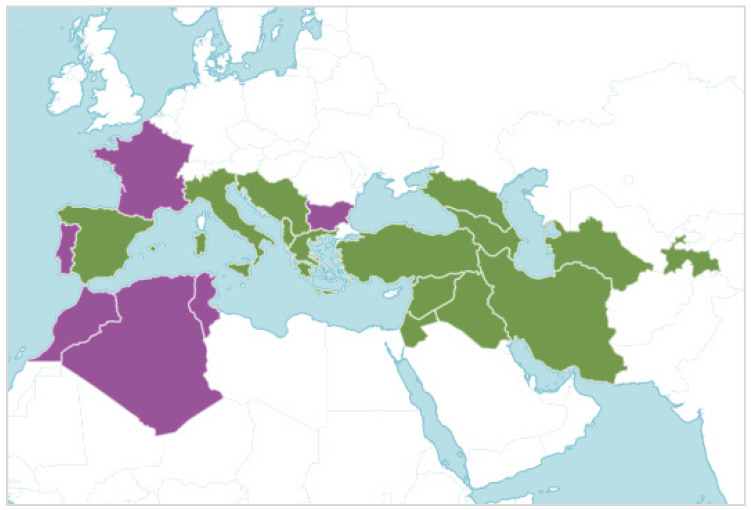
Map showing the distribution of the geophyte *Sternbergia lutea* (L.) Ker Gawl. ex Spreng., which is native of the green areas and introduced to the mauve areas (source: https://powo.science.kew.org, accessed on 16 August 2023).

**Figure 3 plants-12-03043-f003:**
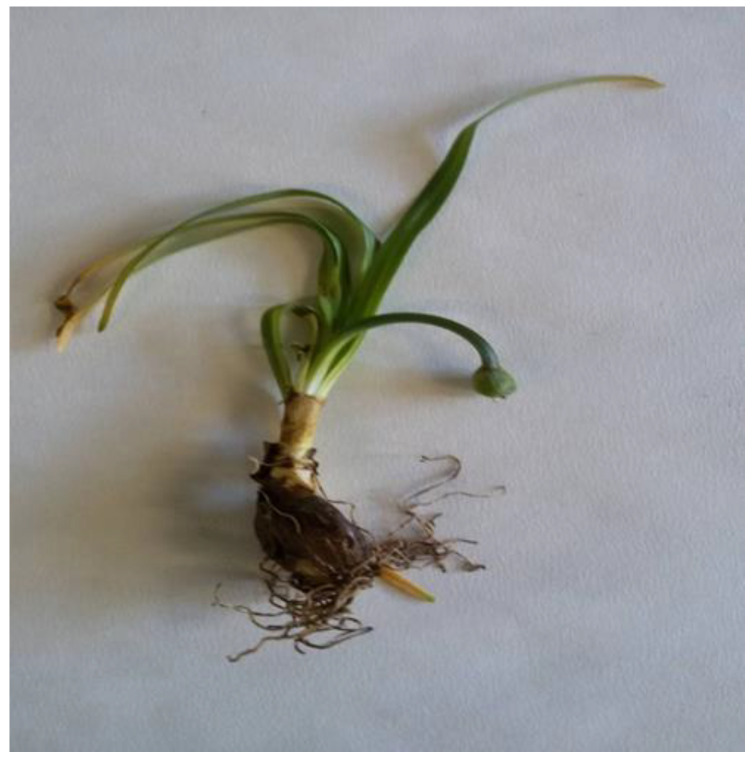
A specimen of *Sternbergia lutea* collected in the field and transferred into the laboratory (Section of Botany, Department of Biology, National and Kapodistrian University of Athens).

**Figure 4 plants-12-03043-f004:**
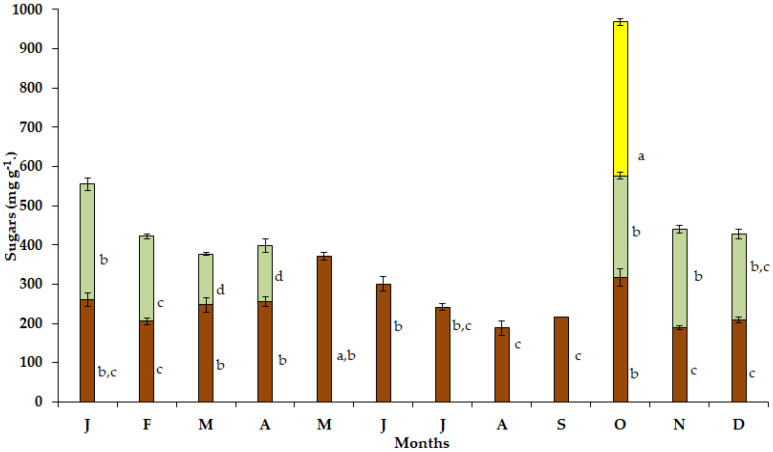
Annual variation in soluble sugars in bulbs (brown bars), leaves (light green bars) and petals (yellow bar) of *Sternbergia lutea* collected from January (J) to December (D). The values are means of three replicates ± standard error. Different lower-case letters indicate statistically significant differences.

**Figure 5 plants-12-03043-f005:**
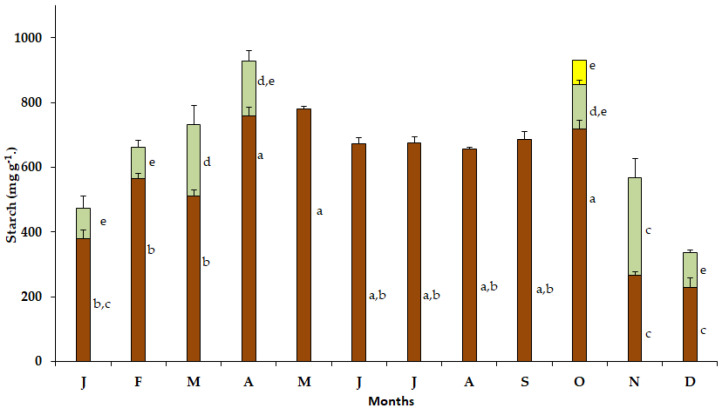
Annual variation in starch content in bulbs (brown bars), leaves (green bars) and petals (yellow bar) of *Sternbergia lutea* collected from January (J) to December (D). Values are means of three replicates ± standard error. Different lower-case letters indicate statistically significant differences.

**Figure 6 plants-12-03043-f006:**
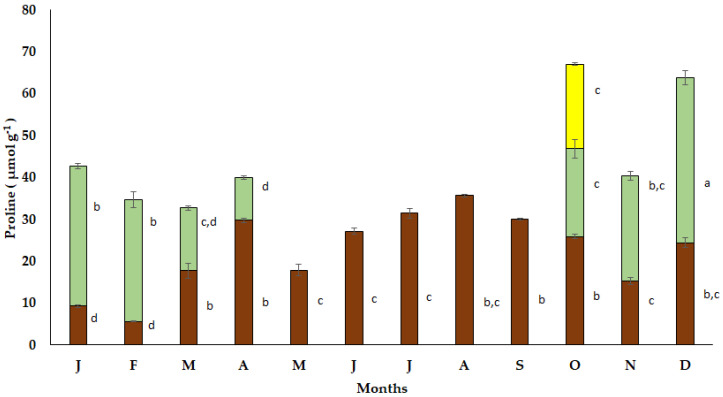
Annual variation in proline accumulation in bulbs (brown bars), leaves (green bars) and petals (yellow bar) of the geophyte *Sternbergia lutea* collected from January (J) to December (D). Monthly values are means of three replicates ± standard error. Different lower-case letters indicate statistically significant differences.

**Table 1 plants-12-03043-t001:** Seasonal water potential (Ψw), osmotic potential (Ψs) and turgor (Ψp) of leaves. The values are means of five replicates ± SE. Significant differences (*p* < 0.05) of mean values are marked using lower-case superscript letters that are given separately on each column variable.

Months	Ψw (MPa)	Ψs (MPa)	Ψp (MPa)
January	−0.50 ± 0.05 ^a,b^	−0.76 ± 0.04 ^a,b^	0.26 ± 0.02 ^b^
February	−0.50 ± 0.02 ^a,b^	−0.70 ± 0.06 ^a^	0.20 ± 0.03 ^c^
March	−0.57 ± 0.04 ^b^	−0.72 ± 0.02 ^a^	0.15 ± 0.01 ^c^
April	−0.67 ± 0.03 ^c^	−0.70 ± 0.04 ^a^	0.03 ± 0.00 ^d^
May			
June			
July			
August			
September			
October	−0.45 ± 0.03 ^a^	−0.71 ± 0.06 ^a^	0.26 ± 0.05 ^b^
November	−0.46 ± 0.02 ^a^	−0.78 ± 0.04 ^b^	0.32 ± 0.03 ^a^
December	−0.45 ± 0.03 ^a^	−0.80 ± 0.05 ^b^	0.35 ± 0.02 ^a^

## Data Availability

The data are available from the authors upon request.

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
