# Peer review of "Unraveling Seasonal Allocation of Soluble Sugars, Starch and Proline in *Sternbergia lutea"

_plants, 2023, doi:10.3390/plants12173043_

Round 1

Author Response

Dear reviewer,

i extend my gratitude for taking time to review my work. Your insightful analysis and thoughtful feedback are invaluable in shaping this work.

Warm regards

Author Response

Dear reviewer,

your feedback was undoubtedly play a pivotal role in shaping the final version and i am committed to delivering a piece that meets the high standards you have set forth.

Warm regards

Reviewer 3 Report

Dear /Editor

Thank you for your invitation to review the paper titled “Unraveling Carbohydrate and Proline Seasonal Allocation in Sternbergia lutea” the authors investigate and compare the partitioning of soluble sugars, starch, and proline between above- and below-ground parts, as well as in floral petals of Sternbergia lutea that are exposed to different environmental conditions. Also, the water status of the leaves of Sternbergia lutea has been investigated.

 I regret to inform you that manuscript not Suitable for Further Processing

General comments:

1-    Manuscript language is poor, use scientific language and significant editing of language is also recommended and a revision of language is also recommended

2-    Currently manuscript look like a thesis rather than a research article. The manuscript should be rewrite.

3-    Results very poor and shortage.

4-    Material and methods good statement and It is the most beautiful thing in the manuscript .

5-    Hypothesis and novelty of study should be elaborated in more detail.

6-      The discussion is very poor, due to the poor results in the manuscript

 Extensive editing of English language required

Author Response

Dear reviewer,

i extend my heartfelt thanks for your time, dedication and the invaluable guidance you have provided.

Warm regards

Round 2

Reviewer 3 Report

the authors improved manuscript  and i am accept it for publish